# Major Emerging Fungal Diseases of Reptiles and Amphibians

**DOI:** 10.3390/pathogens12030429

**Published:** 2023-03-08

**Authors:** Lionel Schilliger, Clément Paillusseau, Camille François, Jesse Bonwitt

**Affiliations:** 1Argos Veterinary Clinic of Paris Auteuil, 35 Rue Leconte de Lisle, 75016 Paris, France; 2SpéNac Referral Center, 100 Boulevard de la Tour Maubourg, 75007 Paris, France; 3Department of Anthropology, Durham University, South Rd., Durham DH1 3LE, UK

**Keywords:** reptile, amphibian, emerging infectious diseases, nannizziomycosis, ophidiomycosis, chytridiomycosis, *Nannizziopsis*, *Ophidiomyces ophidiicola*, *Batrachochytrium dendrobatidis*, *Batrachochytrium salamandrivorans*

## Abstract

Emerging infectious diseases (EIDs) are caused by pathogens that have undergone recent changes in terms of geographic spread, increasing incidence, or expanding host range. In this narrative review, we describe three important fungal EIDs with keratin trophism that are relevant to reptile and amphibian conservation and veterinary practice. *Nannizziopsis* spp. have been mainly described in saurians; infection results in thickened, discolored skin crusting, with eventual progression to deep tissues. Previously only reported in captive populations, it was first described in wild animals in Australia in 2020. *Ophidiomyces ophidiicola* (formely *O. ophiodiicola*) is only known to infect snakes; clinical signs include ulcerating lesions in the cranial, ventral, and pericloacal regions. It has been associated with mortality events in wild populations in North America. *Batrachochytrium* spp. cause ulceration, hyperkeratosis, and erythema in amphibians. They are a major cause of catastrophic amphibian declines worldwide. In general, infection and clinical course are determined by host-related characteristics (e.g., nutritional, metabolic, and immune status), pathogens (e.g., virulence and environmental survival), and environment (e.g., temperature, hygrometry, and water quality). The animal trade is thought to be an important cause of worldwide spread, with global modifications in temperature, hygrometry, and water quality further affecting fungal pathogenicity and host immune response.

## 1. Introduction

Emerging infectious diseases (EIDs) are caused by pathogens that have undergone recent changes in terms of geographic spread, increasing incidence, and expanding host range, or by previously unknown pathogens that are being discovered thanks to advances in surveillance and research, particularly in the field of laboratory diagnostics [1]. Reptiles and amphibians are not protected from the threat of EIDs [2,3,4,5,6,7,8,9,10,11,12]; in recent years, they have been subject to the emergence of bacterial, viral, fungal, and parasitic diseases that are not only increasingly observed in captivity, but are also responsible for wild population declines: arenavirus, nidovirus, paramyxovirus infections, testudinid intranuclear coccidiosis, ophidiomycosis, paranannizziomycosis, nannizziomycosis, and *Emydomyces testavorans* infections (in reptiles) [3,5,10,13,14,15,16], as well as chytridiomycosis (in amphibians) [17,18], cryptosporidiosis, rhabdovirus, adenovirus, iridovirus, ranavirus, and herpesvirus infections (in reptiles and amphibians) [3,5,9,10,18]. This narrative review describes three of the most important emerging fungal diseases of reptiles and amphibians: nannizziomycosis, ophidiomycosis, and chytridiomycosis, as well as how host, pathogen, and environmental characteristics affect the emergence of these diseases.

## 2. Overview of Fungal Pathogens

### 2.1. Reptiles

Most fungal diseases of reptiles were originally grouped under a fungal complex named *Chrysosporium* anamorph of *Nannizziopsis vriesii* (CANV). Many of the pathogens that constituted this group have since been identified. The CANV denomination was abandoned in the 2010s for a new classification, including three genera belonging to the order Onygenales: *Nannizziopsis* spp., *Paranannizziopsis* spp. (Family: Nannizziopsiaceae), and *Ophidiomyces* spp. (Family Onygenaceae) [19,20,21].

*Nannizziopsis* spp. infection in reptiles was formerly known as “yellow fungus disease” (YFD), now replaced by nannizziomycosis (for *Nannizziopsis* spp. infection) and paranannizziomycosis (for *Paranannizziopsis* spp. infection) [22]. Similarly, *Ophidiomyces ophidiicola* (*Oo*) infection in snakes, formerly known as “snake fungal disease” (SFD), has been replaced by ophidiomycosis [22]. SFD is still used to describe a broad set of clinical signs, whereas ophidiomycosis should be restricted to confirmed *Oo* infection. These dermatomycoses can be grouped under the term “onygenalean dermatomycosis” [22]. In this section, we focus on nannizziomycosis and ophidiomycosis as the most common fungal diseases of captive and wild reptiles worldwide [16,22,23].

#### 2.1.1. *Nannizziopsis* spp.

Nannizziomycosis was first described in 1991 in day geckos (*Phelsuma sp*.) and later in 1997 in three different species of captive chameleons (*Calumma parsonii*, *Chamaeleo lateralis*, and *C. jacksoni*), with CANV attributed as the causative agent [24]. The disease was later named YFD after being reported in three captive inland bearded dragons (*Pogona vitticeps*) with deep granulomatous dermatomycosis and yellow discoloration of the epidermis [25]. Nannizziomycosis has since been reported in several species of lizards and crocodilians, and it is now recognized as an emerging disease in both wild and captive animals [26]. Cases of *Nannizziopsis* spp. infection in captive reptiles have been reported in Africa, Asia, Europe, North America, Australia, and New Zealand [19,20,27,28]. The most commonly affected species in captivity are bearded dragons, although other agamids and iguanids are also susceptible [20]. *Nannizziopsis guarroi* is the most frequently reported *Nannizziopsis* species in bearded dragons, but infection with *N. chlamydospora*, *N. draconii*, and *N*. *barbatae* have also been reported as a cause of nannizziomycosis in bearded dragons [27,29,30]. Several cases of cutaneous and systemic infection involving lungs and kidneys attributed to *N. dermatidis* have been reported in chameleons and geckos [27]. *Nannizziopsis crocodili* was first isolated in 1994 and 1997 in saltwater crocodiles (*Crocodylus porosus*) farms; forty-eight hatchlings died in two outbreaks, suggesting a possible age predisposition [31]. More recently, *N. crocodili* was identified from biopsied tissue in a captive juvenile Johnston’s crocodile (*C. johnstoni*) during an outbreak of severe multifocal dermatitis, affecting four of five crocodiles, in which lesions progressed from superficial ulcerations to black pigmentation and localized edema [32]. Of major concern has been the first report of *N. barbatae* in wild animals in 2020, involving four lizard species in Australia, all of which were found dead [28].

Nannizziomycosis begins with initial hyphae proliferation in the outer epidermal stratum corneum, with subsequent invasion of the deeper epidermal strata and dermis. A spectrum of lesions is usually observed, ranging from liquefactive necrosis of the epidermis to granulomatous inflammation in the dermis [33]. In bearded dragons, clinical signs include crusting dermatitis of the face, ventral surface of the limbs, and pericloacal region [30,34,35]. The crusts present with a yellow to brown appearance (Figure 1). Nonspecific clinical signs include molting retention, lethargy, and anorexia. Infection eventually leads to granulomatous inflammation and visceral dissemination, resulting in a poor prognosis [31].

The thermotolerance of reptile-associated *Nannizziopsis* spp. infection is highly dependent on the species involved. *N. chlamydospore*, for example, is moderately inhibited at 35 °C, whereas *N. barbatae* does not grow above 35 °C, and *N. guarroi* grows faster at 35 °C than at 30 °C [20,27,28,31]. Definitive diagnosis of nannizziomycosis is based on both demonstration of fungal elements in affected tissue (via histopathology) and identification of the organism on culture, PCR, or whole genome sequencing [31]. Fungal culture can take up to three weeks, and precise species identification can be challenging.

No standard treatment protocols exist, and the prognosis is guarded because recurrence is commonly encountered. Antifungal susceptibility of reptile-associated *Nannizziopsis* species has been poorly described and is often limited to case reports [31]. Voriconazole and terbinafine show good activity against the main *Nannizziopsis* species, even though resistance is being increasingly described [20,31,36,37,38,39]. Nannizziomycosis treatment consists of wound trimming, systemic appropriate antifungal treatment, and administration of analgesics associated with strict disinfection procedures (materials and enclosure) [20,34,35,37,38]. Animals with confirmed nannizziomycosis or compatible clinical signs should be quarantined and only reintroduced into a captive collection upon a negative PCR and histopathological test. These tests, however, have a low sensitivity in asymptomatic animals, meaning that the introduction of carriers in a naive population cannot be fully excluded. There is a paucity of information regarding environmental disinfection for *Nannizziopsis* sp., but a minimum of two minutes contact with 10% bleach seems to be effective against *N. guarroi* [40]. When handling animals with suspected or confirmed infection, it is recommended to dip gloved hands into disinfectant between handling, allowing sufficient contact time to the disinfectant, followed by rinsing with clean water [41].

#### 2.1.2. *Ophidiomyces ophidiicola*

*Ophidiomyces ophidiicola* (*Oo*) was first described in 2009 as *Chrysosporium ophidiicola* in a black rat snake (*Pantherophis obsoletus*) presenting severe subcutaneous facial swelling, causing displacement of cranial anatomical features [42]. Several cases of snake dermatomycosis were retrospectively identified as ophidiomycosis, including a fatal case in 1990 in wild-caught brown tree snakes (*Boiga irregularis*) imported from Guam to Maryland, United States [43] and free-ranging pygmy rattlesnakes (*Sistrurus catenatus*) in Florida, United States in the 1990s [44]. The presence of *Oo* has been retrospectively demonstrated in museum specimens in the United States as early as 1945 [45,46] and in Europe as early as 1959 [47]. Molecular-based investigations suggest that strains of *Oo* in the eastern United States are primarily represented by four clonally expanded lineages or hybrids between those lineages, and the ancestors of these clonal lineages arrived in the region relatively recently, probably via the animal trade and via human-mediated transmission [11,48]. *Ophidiomyces ophidiicola* is considered responsible for wild population declines of Eastern Massasaugas rattlesnakes (*Sistrurus catenatus*) in Illinois and timber rattlesnakes (*Crotalus horridus*) in Massachusetts, United States [49,50]. Ophidiomycosis is thought to only affect snakes, with cases reported worldwide in colubrid snakes (e.g., *Pantherophis* sp., *Nerodia* sp., *Natrix* sp., *Lampropeltis* sp., *Thamnophis* sp.), viperids (e.g., *Agkistrodon* sp., *Crotalus* sp., *Sistrurus* sp.), Acrochordidae, and boids, elapids, and pythonids [31,51]. Ophidiomycosis is likely overreported in colubrids and viperids in comparison to other species, no doubt because of the important surveillance and research effort in North American snakes; the true number of susceptible species is likely more important than currently reported [4,11,27,46,49,52,53,54,55]. Ophidiomycosis was reported for the first time in Europe in 1985 in a captive ball python (*Python regius*) [27], in captive snakes in Japan [56], and in wild snakes in Hong Kong in 2019 [55] and in Taiwan in 2021 [54]. In North America (including Puerto Rico), *Oo* infection has been reported in at least 49 native snake species and in three non-native species [16].

*Ophidiomyces ophidiicola* is a saprobe with a particular tropism for keratinized environments. Growth is inhibited below 7 °C and above 35 °C, with an optimal temperature of 25 °C [52]. Lesions can be found anywhere along the body, but they initially present as pustular and then crusting dermatitis, involving the face, precorneal scales, thermosensitive dimples, ventral body surface, and the pericloacal region [35] (Figure 2). Regional swelling, edema, and vesiculation may be visible, eventually leading to ulceration. Lesions result in dysecdysis and increased molting frequency. Nonspecific signs include lethargy and anorexia. Infection can progress to underlying tissues including bones, muscle, and viscera. Granulocytic inflammation, edema, and necrosis of the epidermis extending to the dermis are visible on histology. In the wild, ophidiomycosis typically causes a pustular dermatosis in snakes emerging from brumation [4,20,54]. Shedding can reduce or even clear skin lesions [57], which may result in asymptomatic carriers [58,59]. Other fungal infections, such as *Paranannizziopsis* spp., can be confused with *Oo* infection, which can complicate the diagnosis [15,16]. Ophidiomycosis can be classified as possible, apparent, or confirmed, depending on clinical signs, laboratory testing, and demonstration of fungal hyphae on histopathology [15]. Transmission occurs via direct contact with infected individuals or via fomites [53]. Vertical transmission has also been documented [60].

As with nannizziomycosis, the diagnosis of ophidiomycosis is based on both demonstration of tissue involvement and pathogen identification [31]. Medical management is the same as for nannizziomycosis; *Oo* is susceptible to itraconazole, voriconazole, and terbinafine [20,50]. Subcutaneous terbinafine implants with a release over five weeks are being studied for managing venomous snakes [61]. Nebulized terbinafine is also of interest, with therapeutic plasma concentration possibly reached between half an hour and four hours, although efficacy studies are needed [61]. In infections involving captive animals, the environment can be disinfected with common disinfectants, including 3 to 10% bleach or 70% ethanol due to shedding of spores in the environment [41,62]. To avoid contaminating native fauna, snakes should never be released into the wild without first confirming freedom from infection, although this can be challenging because of the relatively low test sensitivity in asymptomatic snakes [58].

### 2.2. Amphibians

#### 2.2.1. Chytridiomycosis

Chytridiomycosis is a fungal disease of amphibians attributed to two pathogenic species of the Chytridiomycete class: *Batrachochytrium dendrobatidis* (*Bd*) and *B. salamandrivorans* (*Bsal*) [63,64,65,66]. Both species differ mainly in terms of their lifecycle (especially temperature and pH requirements), host species, and clinical signs [63,64,67].

*Batrachochytrium* spp. are primitive fungi that inhabit wetlands and aquatic environments [63,64,65,66]. Both species multiply asexually and have an evolutionary cycle consisting of two stages, the motile infectious stage (zoospores) and the immotile reproductive stage (thallus) [66]. The zoospores can move at a speed of about two cm per day in stagnant water, a speed that can be greatly increased in the presence of water currents [68,69]. A second type of non-motile and floating zoospore is produced by *Bsal*. These spores remain infective for over 30 days in pond water and up to 48 h in soil [70]. Following adhesion to the host integument, the zoospore flagellum is resorbed, and a cystic wall is formed [66]. The lifecycle of *Bsal* is complete within five days at 15 °C [71]. The encysted zoospores of *Bd* mature in the zoosporangium and then the thallus for a period of four to five days (at 22 °C in vitro) [72]. A notable difference between the two chytrid species is that *Bsal* continues to divide in the encysted zoospore stage, thereby releasing a large number of zoospores from the thallus [64]. *Bd* is a non-obligate parasite that can survive as a saprobionte in water and moist soil for up to several months [64]. Some isolates of *Bsal* are able to synthesize molecules that enable it to survive as a saprophyte, conferring it the ability to withstand prolonged periods without hosts [73]. *Batrachochytrium* spp. are extremely vulnerable to desiccation.

Amphibian chytrids colonize the keratinized layers of the epidermis (stratum corneum and stratum granulosum) in adults (*Bsal* and *Bd*) and the mouthparts in tadpoles (*Bd* only), thereby disrupting osmoregulation, respiration, and foraging activity [63,66,74]. Once transcutaneous ion exchange is impeded, metabolite imbalances (hyponatremia, hypokalemia, hypochloremia, and hypocalcemia) cause decreased plasma osmolarity, cardiac pathologies, and death [63,64].

Diagnosis is obtained by PCR (or qPCR) of skin swabs or skin biopsies. Cytological examination of skin scrapping or histological examination of skin tissues are also possible, but are less sensitive than PCR [75]. Surveillance in wildlife can be achieved using PCR on environmental DNA (eDNA) [76,77,78,79,80]. Multiple treatment regimens have been described in captivity, including treating water with chloramphenicol malachite green or methylene blue [81], altering the skin microbiota by using probiotics [82,83,84], or increasing the environmental temperature [85,86,87,88,89]. Antifungal therapy (e.g., itraconazole, voriconazole, polymyxin E, or terbinafine) paired with non-steroidal anti-inflammatories are another option [64,65]. Additional microbiological testing can be required to treat superinfections [65]. Zoospores can easily be eliminated by desiccation, UV exposure, heat (4 h at 37 °C), or 5% sodium chloride [90]. Prevention in amphibians destined for trade involves strict quarantine of captive hosts for 60 days with entry and exit PCR testing [91,92]. Outdoor areas are particularly difficult to protect from disease incursion because zoospores can travel in moving water [68] or be transmitted through fomites, for example on the feathers and interdigital skin of aquatic birds [70,93,94] or fomite transmission via anthropogenic activities (e.g., movements of vehicles and equipment, or via footwear) [95,96]. Strict hygiene procedures are therefore required to prevent pathogen translocation [41]; these include cleaning and disinfection of footwear, tires, and other potentially contaminated surfaces [41]. Contact with 70% ethanol (one minute contact time) or 5% bleach (5–15 min contact time) is sufficient to inactivate *Bd* [41]. Non-powdered or vinyl gloves used to handle infected individuals can be disinfected using the same protocol described for *Oo* [41].

##### *Batrachochytrium dendrobatidis* 

*Bd* was first described in 1998 from dead wild anurans collected in Australia in 1993 and Panama in 1994 [97]. Recent studies suggests that East Asia could be the original source of the panzootic lineage [98]. Retrospective investigations of archival samples have revealed its presence in the United States (since 1888) [99], Brazil (since 1894) [100], Asia (since 1902) [101], Africa (since 1933) [102], Canada (since 1961) [103], and Europe (in 1997) [104]. *Bd* has since been reported worldwide in over 1375 species, including in anurans, caudates, and caecilians [64,66,105,106,107]. Molecular investigations suggest that it appears to have been stable in wild populations for many decades, after which it spread globally, most likely as a consequence of the global trade in wild animals [99,108], notably that of the African clawed frog (*Xenopus laevis*) and bullfrog (*Lithobates catesbeianus*) [2,109,110,111,112]. *Bd* is responsible for the decline of amphibian populations around the world, mostly in tropical regions of Africa and South America, but also in Australia and southern Europe [113]. The harlequin frog (*Atelopus varius*), for example, has undergone >90% population declines over the past 10 years, while other species, such as the golden toad (*Incilius periglenes*) and Panamanian golden frog (*Atelopus zeteki*), have gone extinct in the wild [114].

*Bd* can be classified according to lineages, including the very high pathogenic *Bd*-GPL (global pandemic lineage) and endemic lineages, including *Bd*-Cape, *Bd*-Brazil, *Bd*-Asia, and *Bd*-CH [115]. This classification is constantly evolving as new lineages and genotypes are being discovered, including recent evidence of hybridization [116,117].

Growth is inhibited below 10 °C or above 28 °C, with an optimal temperature of 17–25 °C and a pH of 6–7 [64]. In tadpoles, infection is usually limited to the beak, manifested by depigmentation of the mouth and its periphery [97]. This results in decreased food intake and growth, as well as limited swimming ability [74]. In contrast to adults, infection in tadpoles is rarely life-threatening because their integument contains very little keratin. In adults, signs are highly variable and nonspecific. Affected animals may be asymptomatic, lethargic, anorexic, or present with neurological disorders (e.g., abolition of the reversal reflex, ataxia, convulsive, and seizures). Sudden death without overt signs can occur [64]. Dermatologic signs include increased molting frequency associated with hyperkeratosis, hyperplasia (up to 30 times the normal thickness), erythema, and discoloration of the skin [64]. The lesions are typically located on the ventral body surface (mainly on the pelvic patch), hindlimbs, and fingers in anurans (Figure 3) [64,72].

##### *Batrachochytrium salamandrivorans* 

*Bsal* was first described in the Netherlands in 2013 and is responsible for the decline of >99% of wild fire salamanders (*Salamandra salamandra*) in some areas of Europe [63,71,118]. *Bsal* has been detected in captive newts and salamanders in Germany, Spain, and the United Kingdom [6,119,120]. It appears to be limited to European salamanders and newts [12,64,118], while native wild Asian urodeles are suspected to act as asymptomatic reservoirs [118,121,122,123]. A list of amphibian species according to *Bsal* susceptibility (susceptible, asymptomatic carrier, resistant) and geographic location is available elsewhere [124]. *Bsal* chytridiomycosis is, however, not limited to urodele species. Although laboratory studies have demonstrated the inability of *Bsal* to infect caecilians when placed in contact with 10,000 zoospores for 24 h [63,118], common midwife toads (*Alytes obstetricans*) from Europe are susceptible to *Bsal* when exposed to high loads of zoospores (contact with 100,000 zoospores for 24 h), suggesting a potential role of anurans in the pathogen lifecycle [70,125]. Cuban treefrogs can be infected with *Bsal* and, surprisingly, chytridiomycosis can develop in animals at the two highest zoospore dose exposures [126]. Moreover, different strains of *Bsal* might account for variations in susceptible species and epidemic profile, as was hypothesized following the isolation of *Bsal* in wild small-webbed fire-bellied toads (*Bombina microdeladigitora*) from Vietnam [121,127]. Current bans on amphibian transport that largely focus on halting the trade of urodele species may, therefore, be insufficient to prevent translocation of *Bsal*, especially as anurans constitute 99% of global amphibian trade [128].

In urodeles, only the terrestrial life stages are thought to be susceptible. Direct contact for at least eight hours is sufficient for transmission [118], although the delay between exposure and mortality in susceptible species is highly variable, ranging from 12–18 days in fire salamanders to seven weeks in gold-striped salamanders (*Chioglossa lusitanica*) [63,118,127]. In contrast to *Bd*, dermatological lesions present as multifocal ulcerative superficial epidermal lesions that are distributed over the entire body [63,64]. Nonspecific clinical signs (e.g., anorexia, lethargy, and ataxia) and increased molting frequency have also been reported [63,64]. Growth is inhibited below 5 °C or above 25 °C, with an optimal temperature of 10–15 °C and a pH of 6–8 [63,129]. Nevertheless, *Bsal* infections have been reported in *Tylotriton* species at temperatures >26 °C, suggesting a difference in thermotolerance across lineages [123,130].

## 3. Host–Pathogen Relationship

### 3.1. Host-Related Factors

The integument serves as a physical and immunological barrier against pathogens. In most amphibians, the integument is composed of numerous mucus and granular glands responsible for mucus formation. Skin mucus contains various antimicrobial substances, including antimicrobial peptides, lysozymes, antibodies, antifungal proteins, and antifungal symbiotic bacterial communities gathered under the term of microbiome (e.g., *Janthinobacterium lividum*, *Pseudomonas fluorescens*, *and Lysobacter gummosus*), which collectively form the microsome [64,131]. The microsome serves as a major barrier against skin colonization, including *Bd* and *Bsal* [131,132,133]. Under experimental conditions, the microbiome prevents *Batrachochytrium* spp. skin contamination in the first 24 h after exposure by reducing the number of viable zoospores between three to 20 times in *X. laevis* [64]. Symbiotic bacteria secrete several antifungal metabolites that inhibit the growth of *Bd* and are repellent for *Bd* zoospores [64,131,134,135,136]. The production of anti-*Bd* antibodies (IgM, IgX, and IgY) in the microbiome has been demonstrated in *X. laevis*, but their production appears to be inconsistent across species [64,137]. Genes coding for antimicrobial response and bacterial communication present in skin microbiomes have been described in the terrestrial neotropical frog (*Craugastor fitzingeri*) using shotgun metagenomic analysis [138]. Physical damage, or the presence of pollutants in the external environment, can affect the integrity of the microsome. Environmental contamination with microplastics, for example, increases *Bd* pathogen load in midwife toad tadpoles in a dose-dependent relationship [139].

In reptiles, keratin (including beta keratin) prevents infections through mechanical protection. Skin lesion (whether caused by trauma or a primary infection) are a risk factor for epidermal colonization by *Oo*, *Nannizziopsis* spp. and *Paranannizziopsis* spp. [16,20]. Two studies conducted on corn snakes suggest that colonization is less effective in instances of an intact skin barrier [58,140]. In infected animals, molting retention can lead to prolonged contact with the pathogen and increased risk of colonization of deeper tissues [30]. An outbreak of *N. dermatidis* infection in a population of captive veiled chameleons (*Chamaeleo calyptratus*) appears to have been facilitated by molting retention, allowing the development of pathogenic hyphae and colonization of the underlying epidermis [33]. Conversely, the frequency of molting is usually increased in infected snakes, thereby increasing the removal of necrotic tissues and fungal elements, which can lead to clinical recovery, especially if infection is constrained to the superficial epidermis [50,57]. In case of deeper tissue involvement, molting is insufficient to remove the fungus, and it is insufficient to clear infection in a bearded dragon [30]. Increased molting frequency has been described in amphibians with chytridiomycosis, probably as a defense mechanism [64]. A recent study in water snakes (*Nerodia sipedon*) demonstrated that *Oo* infection alters the skin microbiome, leading initially to an increase, and then a decrease in microbial richness compared to control groups, suggesting that disturbances of the host microbiome could affect host susceptibility [141].

In ectotherms, immune activity is directly related to external temperatures. The cellular and humoral responses can be reduced or inhibited at extreme high or low temperatures [142]. This immunological seasonality has only been described in a limited number of species, but is thought to exist at varying levels in most, if not all, ectotherms [142,143,144,145]. A host defense mechanism will, therefore, include heat-seeking behavior to increase immune activity (known as behavioral fever) [146,147,148,149,150]. Behavioral fever has been observed in snakes with ophidiomycosis, as is also described with other infectious diseases of reptiles [4,49,151]. It has also been hypothesized that species adapted to cooler climates could be more susceptible to pathogens when exposed to warmer temperature, and species adapted to warm climates could be more susceptible when exposed to cooler temperature; this hypothesis is named the thermal mismatched hypothesis [152,153,154]. Pathogens, on the other hand, have a relatively broader temperature range tolerance compared to the vertebrate host. According to the thermal mismatch hypothesis, the virulence of a pathogen, therefore, depends on the performance gap between host and pathogen vital rates [154]. This hypothesis had been tested to evaluate the probability of *Bd* infection in two species of caudates using a laboratory experiment; warm-dwelling species had a higher probability of being infected with *Bd* when kept at cool temperatures, but the opposite was not demonstrated, probably because *Bd* is inhibited at warmer temperature [153]. Seasonal cycles in immune response are usually correlated with an increased susceptibility to infectious agents; a potential cause for this is ophidiomycosis in wild ophidians. A laboratory study showed that fungal lesions appear as corn snakes emerge from brumation, at a time when host immunity is decreased (while lesions are rarer when the temperature is higher), consistent with an observed increased prevalence of lesions in wild animals during springtime [140,155,156,157].

Reptiles and amphibians produce cortisol or corticosterone in response to stressors [142,158]. No direct link between blood or skin cortisol levels and the existence or severity of ophidiomycosis has been established to date [140]. In some amphibians (*Alytes* sp.), *Bd* infection induces the production of corticosterone hormones at differing levels according to *Bd* lineages (*Bd*-GPL vs. *Bd*-Cape); the difference in observed cortisol levels could be due to differences in *Bd* pathogenicity [159]. Exposure of tadpoles (*Anaxyrus boreas*, *Rana cascadae*, and *Lithobates catesbeianus*) to exogenous corticosteroids does not influence their susceptibility to *Bd* infection [160]. However, exposure of red-legged salamanders (*Plethodon shermani*) to exogenous corticosteroids did induce a higher infection abundance, but without visibly affecting clinical severity [161]. In a study conducted in spotted salamanders (*Ambystoma maculatum*), exposure to high loads of *Bsal* induced a significant increase in water-borne corticosterone levels and a reduced growth rate [162]. The role of cortisol production in facilitating infection and in the severity of clinical signs, therefore, remains to be determined. Environmental modifications could induce cortisol production facilitating the implantation of the pathogen, or the pathogen itself could induce the production of cortisol by the host.

A negative correlation between the prevalence of ophidiomycosis and low body score (BCS) has been observed, but this remains inconsistent, although snakes with lower BCS at the time of inoculation appear to have a higher mortality rate [140]. Emaciated snakes emerging from brumation are, therefore, at higher risk of infection, although brumation alone is unlikely to be the sole risk factor, and causation remains to be determined. For example, pigmy rattlesnakes (*Sistrurus miliarius*) do not hibernate in some areas; scarcity in the number of prey and decrease in temperatures seem to be responsible for decreased BCS, probably explaining the negative correlation between *Oo* prevalence and BCS [163]. In addition, a separate study in pigmy rattlesnake showed that *Oo* infection was associated with a significant increase in energy maintenance requirement (30–45%) and total evaporative water loss rate (30–40%), regardless of temperature, which might contribute to a reduction in BCS [164]. It therefore remains to be demonstrated whether *Oo* infection results in a decreased BCS, or whether weakened snakes with already low BCS are more susceptible to infection.

### 3.2. Pathogen-Related Factors

Direct contact between the pathogen and the integument is usually required for infection to occur. In the case of *Bd*, flagellate zoospores are able to detect host tissues (keratin derivatives and its main constituents, cysteine, amino acids, mucus, and skin sugars) and actively move through aqueous media [64]. *Bd* has numerous proteases, lipases, and metallopeptidases that compromise skin integrity and allow invasion of host cells [165]. In addition, fungal infection decreases the expression of essential components of local immunity (e.g., keratin, collagen, elastin, fibrinogen, and antimicrobial peptides) and systemic immunity (e.g., reduction of the adaptive response, lymphocytes numbers, and toll-like receptors) of the host, thus promoting the emergence of new sites of infection [64]. Similarly, *Oo* possesses enzymes that can degrade the main elements of the integument (gelatinase, keratinase, and lipase). The production of urease is toxic to host cells and allows further colonization of the fungus [57].

*Bd*-GPL includes the most virulent North and Central American strains associated with wild amphibian population declines [116,166,167]. A study carried out on several strains of *Bd* reveals that susceptibility to *Bd*, in term of prevalence and infectious load, highly depends on both the strain and the amphibian species involved. For example, the survival rate of Western toads (*Anaxyrus boreas*) is not impacted by the Oregon (JEL630) and Maine strains (JEL627), but mortality is high when exposed to the Panama (JEL425) and California strains (JEL646) [168]. In addition, the cohabitation of certain strains could facilitate the appearance of hybrids, which highlights the dangers of the global animal trade as a source for emerging strains. Indeed, a case of hybridization has already been demonstrated between strains of *Bd*-GPL and *Bd*-Brazil [116,117].

Three clades of *Oo* have been identified, including clade 1 (European clade, isolated in the United Kingdom and the Czech Republic), clade 2 (North American clade, isolated in the United States and Taiwan, and clade 3 (isolated in Taiwan and the United Kingdom [11,48,54,169]. Clade 2 appears to have recently emerged, sharing a common origin with the Eurasian clade. It remains to be proven what, if any, differences exist in terms of pathogenicity across clades. The recent introduction of European strains on the American continent via the animal trade is strongly suspected [11,48].

### 3.3. Environment-Related Factors

As described above, *Nannizziopsis* spp., *Oo*, and *Batrachochytrium* spp. have specific temperature requirements for growth. The body temperature of the host animal—and therefore, in ectotherms, of the environment—appears to be a critical factor affecting the pathogen’s ability to cause infection and disease [35]. In one study, corn snakes were inoculated with *Oo* spores, hibernated, and then woken up from brumation and maintained at different temperatures. Animals kept at “springtime” temperatures after brumation had more severe lesions than animals kept at “summertime” temperatures, which might be attributable to a combination of immune system regulation and fungal proliferation [140]. A similar observation was made in wild pygmy rattlesnakes [163], suggesting the existence of a negative correlation between the prevalence of ophidiomycosis and temperature [20,57,163]. Hygrometry also seems to play an inconsistent role in the manifestation of ophidiomycosis, with a negative correlation between hygrometry and the prevalence and severity of lesions observed in pygmy rattlesnakes [163], but positive correlation in wood rattlesnakes [170]. Brumation and lower temperatures seem to be the two main factors explaining the appearance of ophidiomycosis in snakes, although the relative contribution of temperature on fungal proliferation, pathogenicity, and reduced immune function subsequent to brumation remains unclear. In addition, many species of snakes hibernate in community burrows, which facilitates pathogen spread [140,170]. *Oo* is more prevalent in the soil of burrows compared to neighboring soil, and its presence is inversely dependent on soil microbial richness, with increased growth in abiotic soils [171].

The lifecycle of *Nannizziopsis* spp. has been much less studied than those of *Oo* and *Batrachochytrium* spp. In captivity, no predisposing factors for infection have been identified, although poor husbandry conditions have been suspected in infection of two bearded dragons [172]. Although most of the species known to be susceptible to *Nannizziopsis* spp. do not undergo prolonged brumation, short periods of temperatures decreases might but sufficient to allow pathogen infection of the integument of farmed saltwater crocodiles [173]. Data are lacking concerning predisposing factors in the wild; only one description of *Nannizziopsis* sp. outbreak in wild lizard populations in Australia has been documented to date [28]. The lizards were mainly found in moderate to highly urbanized environments, which are often associated with high stress intensity, pathogen burden, and transmission intensity when compared to wildlife populations, all of which could have increased host vulnerability [28,174].

Different factors have been identified as influencing host susceptibility to *Bd* pathogenicity and prevalence, such as season, temperature, altitude, water velocity, and UV-B radiation intensity [64,68,86,90,175,176,177,178]. In many cases, there is a negative correlation between *Bd* infection and environmental temperature >25 °C [64], although, as with *Oo*, the relative contribution of temperature on immune function and pathogenicity is unclear. High-altitude amphibian populations inhabiting cool and humid conditions are at higher risk of infection [64,175]. Climate change projections anticipate that the range of *Bd* will shift to higher and lower latitudes due to an increasingly favorable environmental niche, along with an expansion of areas suitable for the establishment of amphibian hosts in temperate zones of the northern hemisphere [7]. In addition, the spread of *Bd* in tadpoles is greater in species inhabiting flowing water than for those living in standing water, probably due to fungal dispersal in water currents [68]. Furthermore, *Bd* prevalence in montane lakes is negatively correlated with abundance of aquatic microfauna (e.g., *Daphnia* spp., rotifers, and ciliates) and flora (e.g., microalgae) [179,180].

Water quality plays an important role in the prevalence of chytridiomycosis in wild amphibians and infectious load in the environment. In some regions, animals that tested positive by PCR on skin swabs were more likely to be found in waters rich in organic carbon, phosphorus, and total nitrogen (nitrite, nitrate, and ammonium), but of low pH and temperature [181]. The presence of pollutants can modify infection prevalence. A higher number of animals tested positive for *Bd* in waters with high fungicide and low insecticide concentration [181], whereas laboratory and other fields studies revealed an inhibitory effect of some fungicides on the prevalence and growth of *Bd* on the hosts and an inhibitory and lethal effect on *Bsal* zoospores in vitro [182,183]. A study conducted on Pacific treefrog tadpoles (*Hyliola regilla*) demonstrated a protective effect of glyphosate on *Bd* load among exposed larvae probably due to the inactivation of water mold (*Saprolegnia*-*Achlya* clade), which facilitates *Bd* infection [184]. In addition, *Bd* is sensitive to salinity variations, with salinity levels >2 ppt significantly reducing pathogen load, a possible reason why coastal regions act as refuge against *Bd* infection in certain species [185].

## 4. Conclusions

While the global spread of *Oo*, *Bd*, and *Bsal* is being driven by the international pet trade, climate change is expected to provide new opportunities for fungal emergence [11,109,121,186]. Host–pathogen relationships are mediated by the external environment; temperatures, in particular, have a disproportionate influence on fungal pathogenicity and environmental survival, as well as on host immunity in ectotherm. In addition, climate change is expected to affect host distribution, further driving geographic spread [7,154]. Environmental degradation is likely to cause further pressure on host–pathogen relations, although the interactions with climate change are unknown and require urgent attention.

*Batrachochytridium dendrobatidis* and *Batrachochytridium salamandrivorans* have caused catastrophic declines in wild amphibians worldwide, and there is major concern regarding the extent to which emerging lineages will occupy new ecological niches unlocked by climate change [7,108]. Recent reports suggest that the emergence of fungal diseases is also progressing in reptiles. *Nannizziopsis* spp. had only been reported in captive lizards until 2020; detection in symptomatic wild Australian lizards has since been a major cause of concern given the trajectory of global emergence seen with *Oo*. With new pathogenic members of the Onygenales order being discovered, most recently in aquatic turtles in the United States [13,187], surveillance of fungal pathogens of reptiles and amphibians is paramount, and control measures aimed at preventing geographic spread and spillover between captive and wild populations need to be urgently enhanced [106,186].

## Figures and Tables

**Figure 1 pathogens-12-00429-f001:**
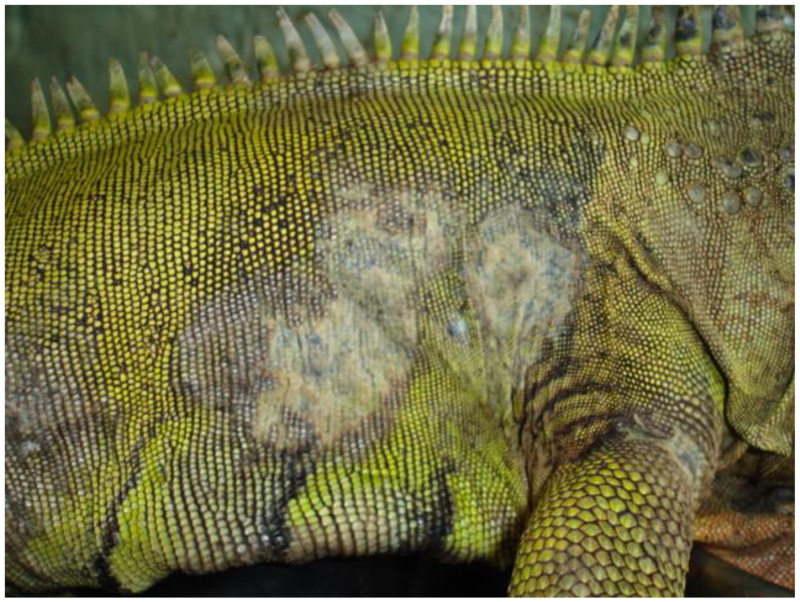
Nannizziomycosis in a green iguana (*Iguana iguana*) presenting with widespread cutaneous lesions of the flank.

**Figure 2 pathogens-12-00429-f002:**
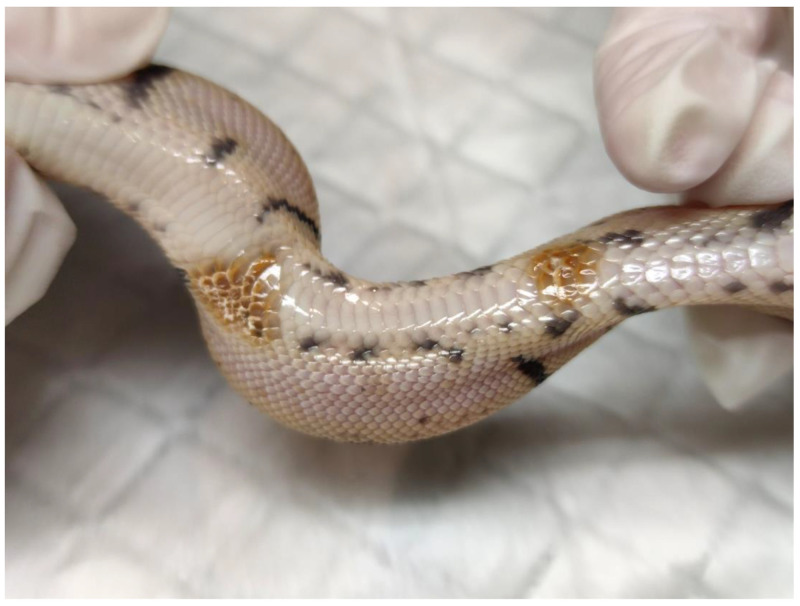
Ophidiomycosis in a ball python (*Python regius*) presenting severe ventral lesions of the epidermis.

**Figure 3 pathogens-12-00429-f003:**
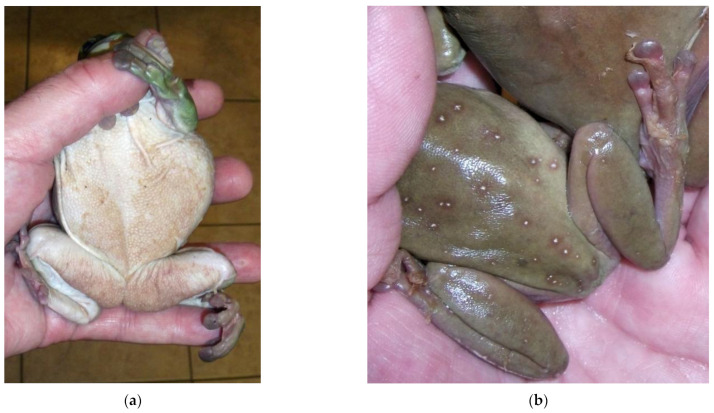
Typical chytridiomycosis lesions in a green tree frog (*Dryopsophus caeruleus*). (**a**) Thickened skin on the ventrum; (**b**) excessive skin shedding on the feet. Photos courtesy K. Wright, In: Mader and Divers (eds). Current Therapy in Reptile Medicine and Surgery, Elsevier, 2014.

## Data Availability

Not applicable.

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
