# Peer review of "Major Emerging Fungal Diseases of Reptiles and Amphibians"

_pathogens, 2023, doi:10.3390/pathogens12030429_

Round 1

Reviewer 1 Report

Please see the attached file with General and Specific Comments.

As I am doing these online forms, I do not see a way to attach pdfs of papers that I suggest you cite. You can email me and I will send you those pdfs:  [email protected]

You can also email me if you have questions about my comments.

Sincerely, Dede

Author Response

Answers to Reviewer 1 in attachment (pdf file).

Reviewer 2 Report

This is a short summary presenting an overview of some of the fungal pathogens considered by some to be emerging in reptiles and amphibians. In the case of the reptile fungi listed, there is some discussion on the “emerging” status based on recent findings, but that is not of central importance. All of the pathogens mentioned are certainly relevant and of interest. While the paper is interesting, it does remain quite superficial, and is not always up to date. Specific issues are commented upon in the attached pdf. However, several general issues are listed here:

One problem is that the authors do not always differentiated between infection and disease, which should be addressed.

It is generally not clear if you are intending to only address animals in captive situations or if you intended to address both wild and captive animals. If the latter, there are a number of things missing from your text.

It would also be helpful when discussing these pathogens and especially the interaction with environmental factors to discuss both the principle of behavioral fever more clearly as well as the thermal mismatch hypothesis. Both are alluded to very vaguely, but not clearly discussed.

For Batrachochytrium spp. It would be of interest to include more information on diagnostics as well as on hygiene, which is an important topic for field studies, private keepers, and the animal trade.

Author Response

Answers to Reviewer 2 in attachment (pdf file).

Reviewer 3 Report

The manuscript entitled “Major Emerging Fungal Diseases of Reptiles and Amphibians” presents a very well written overview of the three most common amphibian and reptile skin emerging infectious diseases.

My major issue with this manuscript is how very reference light it is. There are multiple sentences throughout the manuscript where the authors present some new information that do not have a reference attached to it (e.g. the first two sentences of the introduction have no support, sentence 3-7 in the first paragraph of page 4). While some of the information is quite established, the authors should not expect the reader to trust them blindly.

My second issue refers to section 3. The manuscript does not represent how conflicting information regarding host-pathogen interactions is. In section 3.1 Host-related factors authors completely glance over gene expression differences. In section 3.3 Environment-related factors authors fail to point out that different studies reach different conclusions in relation to the effects of those factors. Just as an example UV has been both claimed to decrease and increase infection (Ortiz-Santaliestra et al., 2011, Hite, 2016).

Line-by-line comments:

Page 9, line 384: Unclear what the authors mean by “whitewater species”

Page 14, line 662-664: reference 105 not cited in text.

Figure 3: Please address the question left by the author as a comment to avoid reader confusion.

Author Response

Answers to Reviewer 3 in attachment (pdf file).
